



# Evolution and dynamics of the vertical temperature profile in an oligotrophic lake

Zvjezdana B. Klaić[1], Karmen Babić[1, 2], Mirko Orlić[1]

[1]Department of Geophysics, Faculty of Science, University of Zagreb, Zagreb, 10000, Croatia
[2]Institute of Meteorology and Climate Research, Karlsruhe Institute of Technology, Eggenstein-Leopoldshafen, Karlsruhe, 76344, Germany

*Correspondence to*: Zvjezdana B. Klaić (zklaic@gfz.hr)

**Abstract.** In this study, the fine-scale responses of a stratified oligotrophic karstic lake (Kozjak, Plitvice Lakes, Croatia; lake fetch is 2.3 km and maximum depth is 46 m) to atmospheric forcings on the lake surface are investigated. Lake temperatures
measured at a resolution of 2 min at 15 depths ranging from 0.2 to 43 m, which were observed during the 6 July – 5 November 2018 period were analyzed. The results show thermocline deepening from 10 m at the beginning, to 16 m at the end of the observational period, where the latter corresponds to approximately one third of the lake depth. The pycnocline followed the same pattern, except that the deepening occurred throughout the entire period approximately 1 m above the thermocline. On average, thermocline deepening was 3–4 cm per day, while the maximum deepening (12.5 cm per day) coincided with the
occurrence of internal seiches. Furthermore, the results indicate three different types of forcings on the lake surface, and two of these forcings have diurnal periodicity: (1) continuous heat fluxes and (2) occasional periodic stronger winds, while the (3) forcing corresponds to occasional nonperiodic stronger winds with along the basin-steady directions. Continuous heat fluxes (1) produced forced diurnal oscillations in the lake temperature within the first 5 meters of the lake throughout the entire observational period. Noncontinuous periodic stronger winds (2) resulted in occasional forced diurnal oscillations in the lake
temperatures at depths from approximately 7 to 20 m. Occasional steady along-the-basin stronger winds (3) triggered both, baroclinic internal seiches with a principal period of 8.0 h, and, barotropic surface seiches with a principal period of 9 min. Lake currents produced by the surface seiches under realistic-topography conditions generated baroclinic oscillations of the thermocline region (at depths of 9–17 m) with periods corresponding to the period of surface seiches (≈ 9 min), which to our knowledge, has not been reported in previous lake studies. Finally, a simple multiple linear regression model of the near-
surface temperature (0.2 m), which depends on the air temperature and wind speed, can only be used as a rough estimate of the daily mean lake temperature under weak wind and undisturbed air temperature pattern conditions.

## 1 Introduction

Two-way interactions between lakes and the atmosphere occur through fluxes in heat, moisture, momentum, and mass (such as carbon dioxide and methane emissions from lakes and the atmospheric deposition of nutrients) at lake surfaces. Recognition
of the importance of these interactions for weather, climate, and lakes has resulted in broadened interest in physical processes





in lakes, which in recent decades has extended from the limnological community to meteorologists and climatologists. Currently, research is facilitated by advancements in observational equipment, modeling techniques and computer resources. Authors have addressed various topics associated with the physical processes in lakes, such as lake stratification modeling (e.g., Stepanenko et al., 2014, 2016; Heiskanen et al., 2015); the improvement of numerical weather and climate models

through coupling with lake models (e.g., Ljungemyr et al., 1996; Balsamo et al., 2012; Xue et al., 2017; Ma et al., 2019); the parameterization of lake-surface fluxes via transfer coefficients (e.g., Xiao et al., 2013); the estimation of the groundwater inflow rates based on observed vertical profiles of water temperature (e.g., Tecklenburg and Blume, 2017); and the influences of lakes on local and/or regional weather, specifically, the influences on surface air temperatures (e.g., Klaić and Kvakić, 2014), wintertime snowstorms produced by lakes (e.g., Kristovich et al., 2017) and lake breezes (e.g., Potes et al., 2017). The

role of lakes in the regional climate has also been investigated (e.g., Bryan et al., 2015), as well as lake responses to climate and land use change (e.g., Huziy and Shushama, 2017; Roberts et al., 2017; Hipsey et al., 2019).

Here, we also focus on lake-atmosphere interactions. The aim of the present study is to investigate the fine-scale evolution and dynamics of the vertical temperature profile driven by forcings on an oligotrophic lake surface. Generally, forcing of lake surface can be periodic or nonperiodic. While periodic forcing can result in periodic variations of lake temperatures (e.g.,

Heiskanen et al., 2015; Lewicki et al., 2016; Frassl et al., 2018; Kishcha et al., 2018; and the present study), establishment of horizontal gradients of lake surface temperature (Filonov, 2002), formation of internal front (bore) (Filonov et al., 2006) or other nonlinear phenomena associated with energy dissipation (e.g., Boegman and Ivey, 2012), and/or internal oscillations in resonance with periodic winds (e.g., Vidal et al., 2007; Vidal and Casamitjana, 2008), nonperiodic forcing produces surface and internal seiches, which are in more detail described in Section 4.4.

Some authors report on observed high-frequency (frequencies above 0.1 cycle per min, that is, periods below ≈ 10 min) oscillations in thermocline regions of stratified lakes. Thorpe et al. (1996) discussed internal waves with periods between 6 and 10 min which they observed in Lake Geneva within a few hours after the passing of disturbance which caused rapid jump in the thermocline depth. Authors proposed three possible sources of these oscillations: (1) a soliton packet following the jump, (2) waves generated as the jump passed over or around rough topography, and (3) a moving disturbance produced by

thermocline jump near the shoreline accompanied with a region of high shear, low Richardson number and wake-like pattern of radiating internal waves. Stevens (1999) described waves with periods of ≈ 2–10 min in a small, stratified lake (3.9 km long, maximum depth of 65 m). These waves occurred more readily in the middle of the along-the-lake axis and they were short in duration, and according to the author, they occurred due to the basin-scale wave-driven shear associated with internal seiches. High-frequency oscillations in small to medium sized lakes (where Coriolis force can be neglected) have also been investigated

in modelling (Horn et al., 2001; Dorostkar and Boegman 2013) and laboratory studies (Boegman et al., 2005, 2005a; Boegman and Ivey, 2012). Common to all of these lake studies is that they associate high-frequency thermocline oscillations with the degeneration of initial basin-scale internal waves, that is, with a downscale energy transfer from basin- to smaller scales. For small to medium sized lakes (i.e., for mid-latitude lakes with a lake width less than approximately 4–5 km), Horn et al. (2001) identified five different mechanisms of degeneration: (1) damped linear waves, (2) solitons, (3) supercritical flow, (4) Kelvin-



Helmholtz billows, and (5) bores and billows. In contrast to these studies, here we show that observed high-frequency internal oscillations can also be produced by surface seiches.

For this purpose, we analyze the vertical profiles of the lake temperatures observed at a very fine temporal resolution (2 min) at 15 depths ranging from near surface to near bottom during the lake stratification period. The results are presented for one of

the Plitvice Lakes in Croatia, and the observational lake depths are between 0.2 and 43 m.

The Plitvice Lakes are a karstic lake system situated in an inland mountainous area. This system is an approximately 9-km-long chain of 16 lakes that are interconnected by cascades and waterfalls. The lakes descend from south to north from 637 to 475 m above sea level (e.g., Meaški, 2011). These lakes are dimictic (e.g., Špoljar et al., 2007), and apart from the uppermost lake (Prošće), the lakes are oligotrophic (Petrik, 1958, 1961). The uniqueness of the entire lake system is in the creation and

growth of tufa barriers that separate the lake chain into individual lakes. For the Plitvice Lakes, this process is very vulnerable since it occurs within a narrow range of the physical, chemical and biological conditions (Pevalek, 1935; Kempe and Emeis, 1985; Emeis et al., 1987; Chafetz et al., 1994; Horvatinčić et al., 2000; Frančišković-Bilinski et al., 2004; Matoničkin Kepčija et al., 2006; Sironić et al., 2017). Therefore, monitoring the limnological variables and understanding the processes occurring within the lakes is of the utmost importance for preservation of this unique lake system. However, physical conditions and

processes within the Plitvice Lakes have been poorly investigated (Klaić et al., 2018). Few studies have reported on the observed physical properties of lakes, such as lakes temperatures, water conductivity and transparency (Gligora Udovič et al., 2017; Sironić et al., 2017). Some authors report on lake stratifications based only on occasional observations of temperature profiles (Gavazzi, 1919; Petrik, 1958, 1961), while others investigated surface seiches (Gavazzi, 1919; Pasarić and Slaviček, 2016) and mentioned two possible events of internal seiches seen as vertical movements of the isothermal surfaces (Gavazzi,

20 1919).

We address the physical processes within Kozjak Lake (Fig. 1) during its stratification period in 2018. With an altitude of 535 m above sea level (ASL), Kozjak Lake is the twelfth lake in the chain and is both the deepest (maximum and average depths of 46 and 17.3 m, respectively) and the largest lake (lake surface area of 0.82 km$^2$, volume of 0.01271 km$^3$) (e.g., Babinka, 2007; Gligora Udovič et al., 2017). The lake stretches from south-southeast to north-northwest and has a complex

morphometry. The downstream length of the lake (which is not a straight line) is 3095 m, while the along-the-basin axis length, that is, the maximum lake fetch, is approximately 2300 m long. The lake width varies from approximately 100 to 600 m. In addition, in the southern part of the lake, an islet (35–50 m wide and ≈ 250 m long) stretches in roughly the along-lake direction. Moreover, approximately 700 m downstream from the northern tip of the islet, there is a north-south stretching submerged barrier (Gavazzi, 1919). The top of the barrier is approximately 5–6 m below the lake surface. In the past, the barrier divided

the current lake into two separate lakes, whereas at present, this barrier divides the lake into two sub-basins: a deeper sub-basin (northern, sub-basin fetch ≈ 730 m) and a shallower sub-basin (southern, sub-basin fetch ≈ 1570 m).





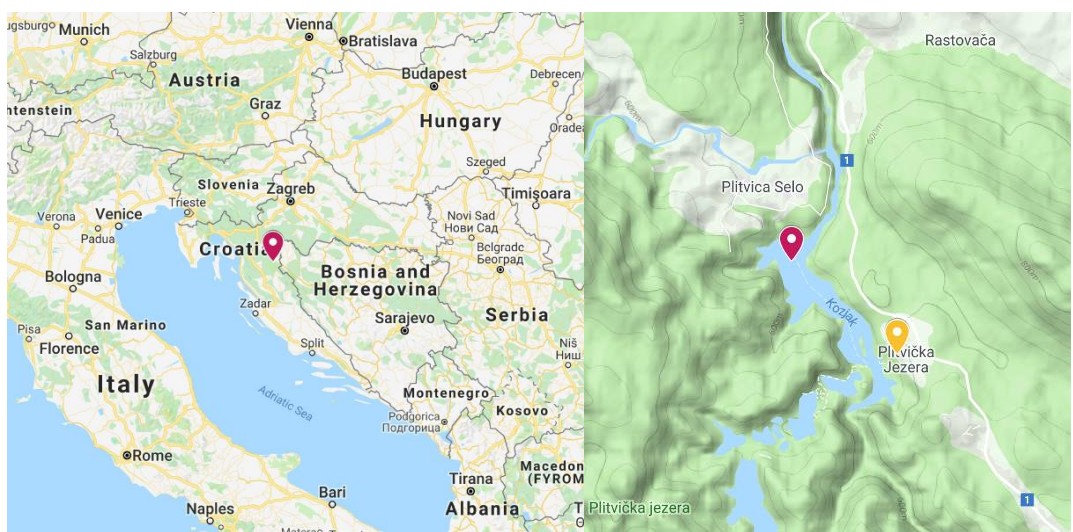

**Figure 1: Position of Kozjak Lake (red bubble, left). A closer look at Kozjak Lake (right). Position of the lake temperature measuring point ($\varphi$ = 44.8902°N, $\lambda$ = 15.6038°E, height of the lake surface 535 m ASL) and meteorological measuring site ($\varphi$ = 44.8811°N, $\lambda$ = 15.6197°E, 579 m ASL) are shown with red and yellow bubbles, respectively. (Source: © Google Maps)**

## 2 Experimental data

### 2.1 Lake temperatures

Lake temperatures were measured with waterproof temperature sensors (HOBO TidBit 400), which are equipped with data loggers. The measurement accuracy is ±0.20°C and ±0.25°C for positive (between 0° and 70°C) and negative (between -20° and 0°C) temperatures, respectively. The sensors measure temperature every second, while the averaging interval of the stored data is specified by a user. In the present study we stored the 2-min means.

Fifteen factory calibrated sensors were fastened to a string at fixed depths ranging from 0.2 to 43 m (specifically, at 0.2, 0.5, 1, 1.5, 3, 5, 7, 9, 11, 13, 15, 17, 20, 23, and 43 m). The string was attached to a buoy that was moored to ensure its fixed position in the deepest part (46 m) of the lake ($\varphi$ = 44.8902°N, $\lambda$ = 15.6038°E; Figure 1, right). Lake temperatures were recorded continuously during the period from 6 July 2018 at 18:00 h local standard time (LST; without summertime advancement by one hour) to 5 November 2018 at 11:24 LST.

### 2.2 Meteorological data

Meteorological data were taken from the automatic meteorological station Plitvička Jezera ($\varphi$ = 44.8811°N, $\lambda$ = 15.6197°E, altitude 579 m ASL, Figure 1, right). The station is maintained by the Croatian Meteorological and Hydrological Service, and the same service also performs quality control of the measured data. Hourly means of the surface (2 m above the ground) air temperature, air pressure, and relative humidity, together with the hourly precipitation amount and surface (10 m above the ground) wind speed and wind direction were available for the period concurrent with the lake temperature observations.


## 3 Methods

In addition to the standard statistical procedures, the methods described in the subsections below were also applied.

### 3.1 Calculation of thermocline and pycnocline depth

Thermocline and pycnocline depths were calculated by a fitting procedure designed for a two-layer model (Krajcar, 1993;

Krajcar and Orlić, 1995). Specifically, a step-like function was fitted to an empirical vertical profile of a scalar variable $s$ by the least square method. The variable $s$ can generally be any scalar, such as, water temperature or density, and its vertical profile is given by a set of $N + 1$ equidistant values of $s$. If an initial empirical vertical profile is not equidistant, the corresponding equidistant profile can be produced by linear interpolation. The two layers are separated by a discontinuity in the vertical profile (e.g., thermocline or pycnocline), which is found at the $h^{th}$ depth.

We denote $N + 1$ equidistant values of $s$ with $s_i$, where $i = 0, 1,\ldots\ldots, N$, and, $s_0$ and $s_N$ correspond to the surface and bottom values, respectively. Furthermore, we assume the following:

$$s_i = A_h \qquad \text{for } i = 0, 1,\ldots, h,$$

and    (1)

$$s_i = B_h \qquad \text{for } i = h+1, h+2,\ldots, N.$$

The appropriate values of $h$, $A_h$ and $B_h$ result in a minimum value of $C_h$, where:

$$C_h = \sum_{i=0}^{h}(s_i - A_h)^2 + \sum_{i=0}^{h}(s_i - B_h)^2,$$

and

$$A_h = \frac{1}{h+1}\sum_{i=0}^{h} s_i,$$    (2)

$$B_h = \frac{1}{N-h}\sum_{i=h+1}^{N} s_i.$$

$C_h$ should be calculated for every $h$, and subsequently, the $h$ that minimizes $C_h$ should be selected. Calculations can be simplified using the following recursive formulas:

$$A_{h+1} = \frac{h+1}{h+2}A_h + \frac{s_{h+1}}{h+2},$$

$$B_{h+1} = \frac{N-h}{N-h-1}B_h - \frac{s_{h+1}}{N-h-1},$$    (3)

$$C_{h+1} = C_h + (s_{h+1} - A_h)^2\frac{h+1}{h+2} - (s_{h+1} - B_h)^2\frac{N-h}{N-h-1},$$

where the initial values of $A$, $B$ and $C$ are as follows:

$$A_0 = s_0,$$

$$B_0 = \frac{1}{N}\sum_{i=1}^{N} s_i,$$    (4)

$$C_0 = \sum_{i=1}^{N}(s_i - B_0)^2.$$

Recursive Eqs. (3) can generally be applied only if $h < N - 1$, since the $h = N - 1$ denominators in terms of $B_{h+1}$ and $C_{h+1}$ (that

is, $N - h - 1$) are equal to zero. Therefore, in the present study, the values obtained from Eqs. (3) were calculated for an $h$





increasing from $h = 1$ to $h = N - 2$. However, we note that it does not affect the result (thermocline or pycnocline depth) if the lowermost layer (that is, the $N^{th}$ layer) is far below the thermocline/pycnocline.

The thermocline depth was determined directly from the measured lake temperatures using the above procedure, and prior to the pycnocline depth calculation, the water density $\rho$ (kg m$^{-3}$) was computed from the formula for freshwater (e.g., Sun et al.,

5     2007):

$$\rho = (1 - 1.9549 \cdot 10^{-5} |T - 277|^{1.68}) \cdot 10^3, \tag{5}$$

where $T$ is the observed water temperature (K). The above formula does not consider the influence of the total suspended sediment on water density, which is a common approximation in freshwater investigations (e.g., Ji, 2008). For both water temperature and water density, individual equidistant vertical profiles with vertical spacings of 0.25 m were generated by linear

interpolation, where the top and bottom depths corresponded to 1 and 43 m, respectively.

## 3.2 Spectral analysis

To assess the frequency content in investigated data sequences (in the present study, we inspect the time series), a spectral analysis, which decomposes the data into a sum of weighted sinusoids, was applied. Both limnological and meteorological

data are expected to contain random effects (noise), which can obscure the investigated phenomenon. In other words, they are not deterministic. Therefore, the Fourier transform computation is not applicable. Instead, a power spectral density (PSD), which shows how the frequency content of the investigated data sequence varies with frequency and is appropriate for the analysis of data containing noise, should be calculated (e.g., Solomon Jr., 1991). In the present study, we determined PSDs using Welch's method (Welch, 1967). This method is also known as the weighted overlapped segment averaging (WOSA)

method or periodogram averaging method.

The WOSA method consists of several steps. First, the initial time series $x[0], x[1],….,x[N-1]$ is partitioned into $K$ segments or batches, where $x[0], x[1],…., x[M-1]$ is the first segment, $x[S], x[S+1],…., x[M+S-1]$ is the second segment, and so on. The final $K^{th}$ segment is $x[N-M], x[N-M+1],…., x[N-1]$, where $M$ and $S$ are the number of points in each segment (i.e., batch size) and number of points to shift between segments, respectively. The parameter $S$, which is usually in the range

of $0.4M \leq S \leq M$ (Solomon Jr., 1991), controls how much the two adjoining segments overlap. For example, if $S = M$, the two segments do not overlap, while for $S = 2$ two adjoining segments overlap in $M - 2$ points.

The second step is to compute a windowed discrete Fourier transform $X_k(v)$ for each segment (from $k = 1$ to $k = K$) at some frequency $v = i / M$, where $-(M/2-1) \leq i \leq M/2$:

$$X_k(v) = \sum_m x[m]w[m]exp(-2j\pi vm), \tag{6}$$

$m = (k-1)S,…., M+(k-1)S-1$ and $w[m]$ is the window function. Furthermore, for each segment, a modified periodogram value $P_k(v)$ is calculated from the discrete Fourier transform:

$$P_k(v) = \frac{1}{W}|X_k(v)|^2, \tag{7}$$





where $W = \sum_{m=0}^{M} w^2[m]$.

Finally, Welch's estimate of the PSD is obtained by averaging the periodogram values:

$$S_x(\nu) = \frac{1}{K}\sum_{k=1}^{K} P_k(\nu). \qquad (8)$$

In the PSD calculations, we used MATLAB software (Version R2018a), which has a built-in function, *pwelch* for estimation

of the PSD of the input signal using Welch's method. Each input time series (which has a total of *N* consecutive data) was split

into $K = 8$ segments of equal length with 50% overlap (i.e., $S = M/2$). The number of points in each segment (*M*) for a particular

time series was controlled by the number of points in the input time series (N) and condition $K = 8$. The remaining (trailing)

input *x* values that could not be included in the eight segments of equal length were discarded. Furthermore, each segment was

windowed with a Hamming window (e.g., Solomon Jr., 1991; Oppenheim et al., 1999; Patel et al., 2013), where the window

function is $w[m] = 0.54 - 0.46\cos[2\pi m/(L-1)]$ for $0 \le m \le L-1$; otherwise, $w[m] = 0$. The window length *WL* in the window

function is set to $WL = 512$ points in present study.

## 4 Results and discussion

Figure 2 shows the observed lake temperatures and atmospheric data during the period from 7 July 2018 at 00:00 LST to 5

November 2018 at 00:00 LST.  As expected, the diurnal variation is clearly seen in the air temperature, wind speed and relative

humidity except for days with synoptically disturbed weather conditions. For example, such disturbed conditions are found on

several occasions at the end of the observational period (as of 25 October, Figure 2). Specifically, during approximately the

last 10 days, winds, which were predominately southeastern (wind direction ≈ 135 deg), were stronger than at other times

(Figure 2c). The air pressure first started to decrease, and eventually, on 30 October, it reached the minimum value, and it

subsequently increased again (Figure 2e). This 10 day period was also accompanied by disturbed diurnal variations in the air

temperature and relative humidity (Figures 2b and 2g) and occasional precipitation (Figure 2f).

The lake temperature in the upper layers also exhibited diurnal variations (Figure 2a), which can be seen with the naked eye

even down to a depth of 5 m. Not surprisingly, the temperature amplitude gradually decreased with depth. During July–

September, at a depth of 0.2 m the temperature amplitude was between approximately 1 and 3.5°C, while at 5 m, it was mainly

below 0.5°C.

Finally, a comparison of upper layer lake temperatures with concurrent wind speeds shows that the downward mixing of lake

layers (and consequent deepening of the mixed layer) coincided with elevated wind speeds, where stronger winds were mainly

southeastern. For example, strong southeastern winds on 26 October (Figures 2c and 2d) resulted in mixing of the uppermost

13-m-deep lake column (Figure 2a). Moreover, synoptically induced high-wind atmospheric disturbances also produced

prominent oscillations in the lake temperature at a depth of 15 m as of 29 October. These oscillations will be investigated in

more detail in Section 4.4.



Figure 2: Observed 2-min mean lake temperatures at various depths from 0.2 to 43 m during the period of 7 July–5 November 2018 (panel a) and simultaneous hourly values of the air temperature (b), wind speed (c) and direction (d), air pressure (e), precipitation amount (f), and relative humidity (g). Measuring point positions are depicted in Figure 1, right.


## 4.1 Thermocline and pycnocline evolution

Figure 3 shows the observed 2-min mean lake temperatures and corresponding water densities calculated from Eq. 5. (panels a and b, respectively) as well as the vertical gradients of the temperature and density (panels c and d) during observational period. Vertical gradients were calculated for each layer bounded by the two adjacent measurement depths as the difference

between the temperature (density) at the depth $d_{n+1}$ and the temperature (density) at the depth $d_n$ divided by the vertical distance between these two depths (that is, $d_{n+1} - d_n$), where $n = 1,…,$ N-1, and N is the total number of measurement depths. Here, N = 15, and $d_1$ and $d_{15}$ correspond to 0.2 and 43 m, respectively. That is, the $z$-coordinate was oriented from the lake surface toward the lake bottom. Furthermore, it is assumed that the vertical gradients are constant throughout entire lake layer placed between the two adjacent measurement depths. The positions of the thermocline and pycnocline, which are also shown in Fig.

3, were determined by the fitting procedure described in Section 3.1.

At the beginning of the observational period (7 July 2018), the lake is already stratified. The lake temperature decreases from 20.1°C (Fig. 3a, top of the lake) to 4.2°C (bottom of the lake), while the water density simultaneously increases from 997.9 kg m$^{-3}$ (Fig. 3b, lake top) to 999.5 kg m$^{-3}$ (lake bottom). Vertical temperature gradients are the highest in first 0.5 m of the lake where occasionally, values of $\Delta T/\Delta z \approx$ -7°C m$^{-1}$ are observed, while in the layer between 0.5 and 1 m their highest observed

magnitudes correspond to $\Delta T/\Delta z \approx$ -3°C m$^{-1}$. The latter is similar to magnitudes reported by Filonov (2002) for the first 1–2 m of a large, but shallower subtropical Lake Chapala. As expected, vertical temperature gradients in the epilimnion of Kozjak Lake are noticeably higher than those observed for large lakes (e.g., Zavialov et al., 2018).

The thermocline and the pycnocline are between 10.50 and 10.75 m (panels a and c) and between 9.25 and 9.50 m (b and d), respectively. Over time, the thermocline and pycnocline both deepened, and by the end of the observational period (5

November 2018), when the lake stratification was already weakened, the thermocline and pycnocline were found at depths of 16.00 and 15.00 m, respectively. During the investigation period, the average rate of deepening of the thermocline (pycnocline) was approximately 1.1 m per month (that is, 3–4 cm per day), while the most intense deepening (approximately 12.5 cm per day) of both the thermocline and pycnocline was between 25 September and 7 October 2018 and 23 October and 2 November, respectively. Furthermore, as seen from Fig. 3c, the strongest lake stratification occured between 9 August and 4 October,

when the temperature in the thermocline region decreased with depth by approximately 3°C per meter (i.e., $\Delta T/\Delta z$ = -3°C m$^{-1}$) except for 25 September, when a weaker vertical temperature gradient (i.e., $\Delta T/\Delta z \approx$ -2°C m$^{-1}$) was found. This period of increased vertical temperature gradient in the thermocline region was accompanied by an increase in water density of approximately 0.4 kg m$^{-3}$ per meter of depth in the pycnocline region. Finally, we note, that although the thermocline and pycnocline were quite similar during the observational period, they did not coincide. This is because the relationship between the water temperature and water density (Eq. 5) is non-linear. A comparison of the thermocline (panel a or c) and pycnocline

depths (panel b or d) shows that the thermocline depth was approximately 0.50–1.75 m greater than the pycnocline depth. The departure of the thermocline from the pycnocline was the largest at the beginning of the observational period (1.00–1.75 m), and it decreased toward the end of the period (0.50–1.25 m), while this difference was 1.03 m on the average.

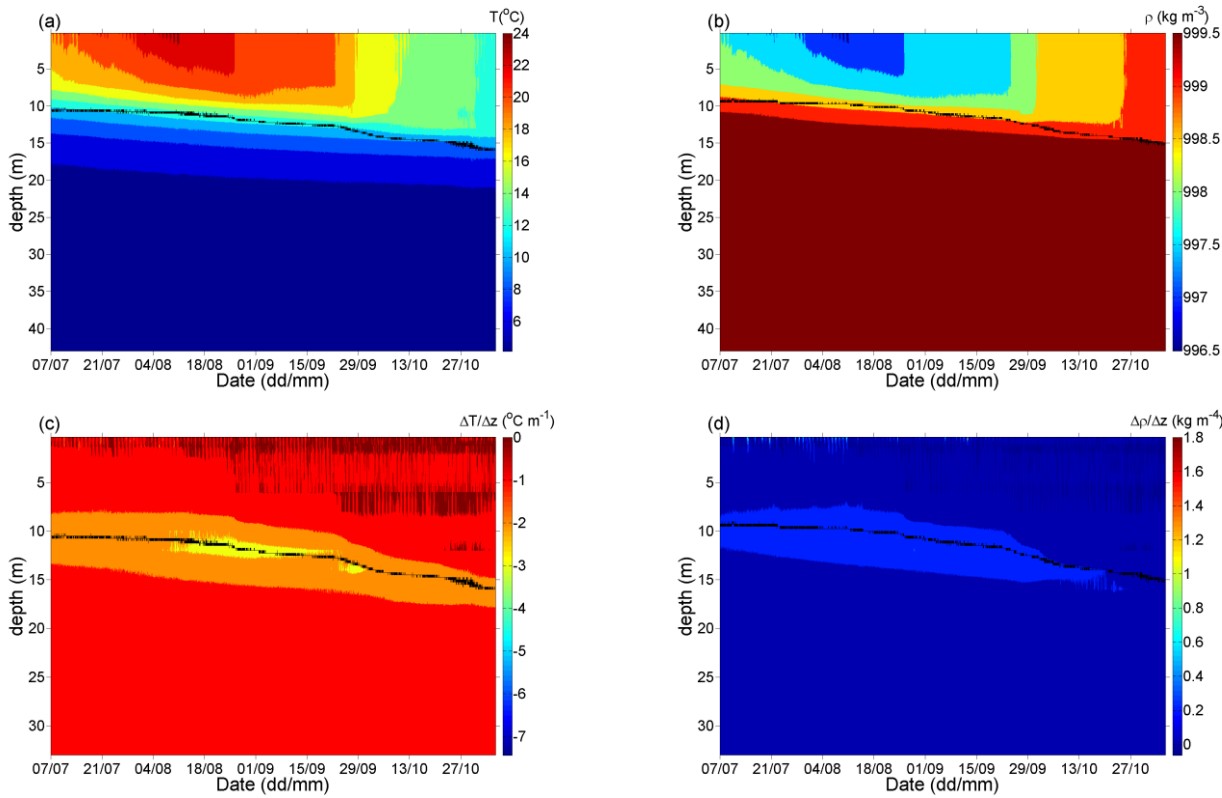

**Figure 3: Observed lake temperatures during the period of 7 July–5 November 2018 (a) and corresponding water densities calculated**

**from Eq. (5) (b). Panels c and d depict the vertical gradients of water temperature and density. The black lines in panels a and c**

**show the thermocline depth, while those shown in panels b and d depict the pycnocline depth. Both, the thermocline and pycnocline**

**depths are determined as described in Section 3.1. The temporal resolution of all data is 2 min.**

Brunt-Väisälä frequency (buoyancy frequency $N$), where $N^2 = (-g / \rho)(\partial \rho / \partial z)$ (e.g., Sun et al., 2007; Boehrer and Schultze,

2008), indicates the maximum frequency for internal waves that can propagate in respective stratification (e.g., Boehrer

and Schultze, 2008). The results for $N^2$ (not shown here) are very similar to those for the vertical gradient of the water density

shown in Fig. 3d. This result is expected since $N^2$ depends on the vertical gradient of water density. During the investigation

period, $N^2$ was approximately zero over the entire lake depth, except for the two regions where it was somewhat higher. One

region is the pycnocline/thermocline (where internal waves may occur), where $N^2$ was approximately $3 \cdot 10^{-3}$ s$^{-2}$, and the other

region is the uppermost 1–2 m of the lake, where during the daytime, it was up to $4$–$16 \cdot 10^{-3}$ s$^{-2}$. The values of $N^2 \approx 3 \cdot 10^{-3}$ s$^{-2}$

suggest that in Kozjak Lake, internal waves with periods as low as 18 s can occur.



## 4.2 Diurnal variations

Figure 2a shows that the diurnal variation in the lake temperature can be seen by the naked eye in approximately the uppermost 5 m of the lake. As expected, the daily temperature amplitude was the highest at the 0.2 m depth (where for some days, it was above 3°C), and the amplitude gradually decreased with depth. The same is also observed in the pattern of the average diurnal

variation in lake temperature within the first several meters of the lake (Figure 4c). Additionally, a comparison of the average diurnal variation in the air and lake temperatures (panels 4a and 4c) reveals a delay of approximately 2 hours in the lake surface (first few meters of the lake) maximum temperature with respect to the maximum air temperature observed at 2 m above the ground. Similar behavior is also found for the daily course of the water density (panel 4b) and vertical gradient of the water density (panel 4d), where a delay between the maximum air temperature and both, the minimum water density and maximum

vertical gradient is also approximately 2 hours. Furthermore, daily courses of lake temperature (4c) and water density (4b) show that during the warm part of the year conditions favorable for the vertical mixing of the uppermost 1–2 m of the lake are established at 6 LST on the average, that is, at approximately sunrise. Namely, at approximately sunrise, a parcel of colder/denser water is found above the warmer/less dense water.

Vertical gradients in the water density within the first few tens of cm of the lake occurred during the afternoon hours up to

approximately 0.15 kg m$^{-4}$ on average (panel 4d). At depths below ≈ 4 m, diurnal variations are not observed. However, the magnitudes increased with depth (not shown here) from approximately 0.06 kg m$^{-4}$ (at 6 m) to approximately 0.22 kg m$^{-4}$ (pycnocline region, at approximately 11 m on average). Below the pycnocline, the vertical gradients of the water density again decreased with depth. Thus, at depths below 20 m, the vertical gradients of the water density were below 0.02 kg m$^{-4}$. As expected, the daily course of $N^2$ (not shown here) exhibited the same pattern as the vertical gradients of water density. During

the afternoon hours, within the first few tens of cm of the lake, values up to approximately $1.4 \cdot 10^{-3}$ s$^{-2}$ are found, while in the pycnocline region, they were up to approximately $2.3 \cdot 10^{-3}$ s$^{-2}$.





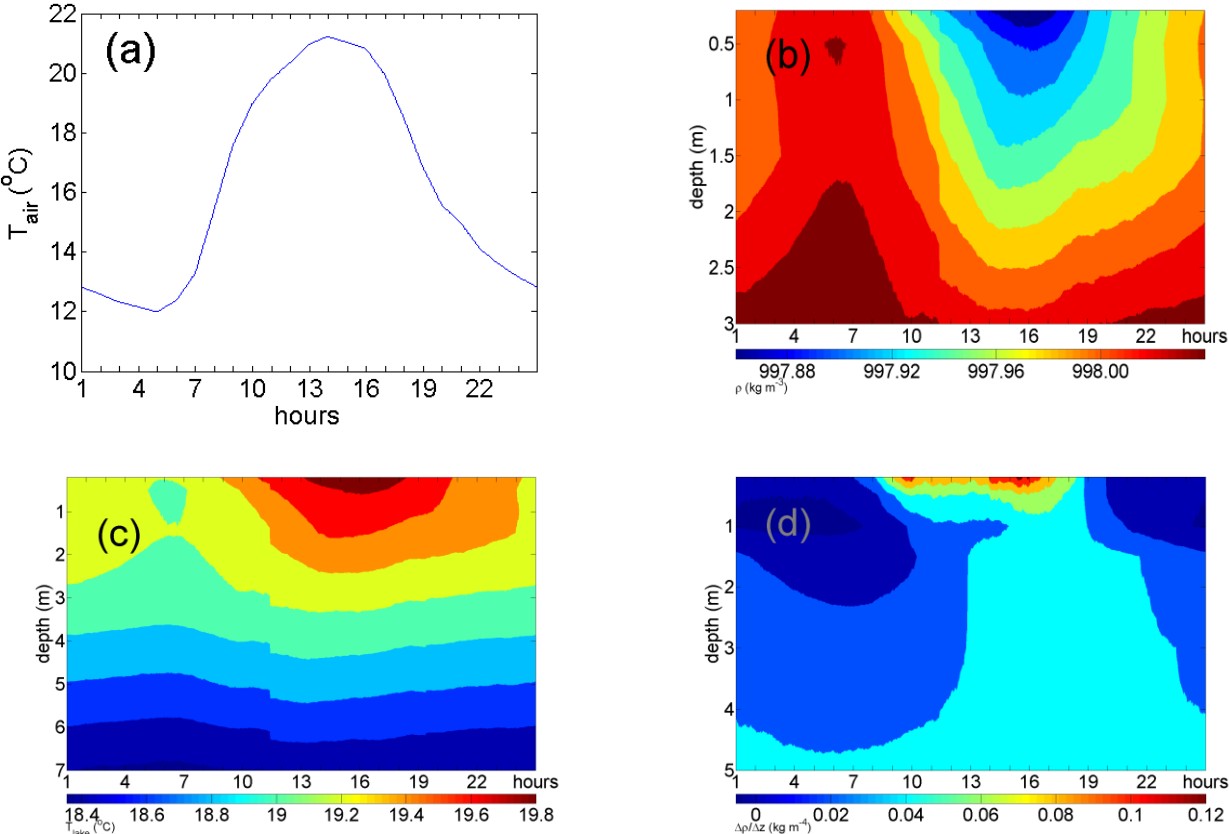

**Figure 4: Diurnal variations in the observed air (2 m above the ground) and lake temperatures during the period of 7 July–5 November 2018 (panels a and c, respectively). Panels b and d depict the diurnal variations in water density (kg m$^{-3}$) and the vertical gradient of the water density (kg m$^{-4}$), respectively. The time at the abscissa corresponds to LST.**

## 4.3 Spectra

Figures 5 and 6 illustrate some of the spectra computed as described in Section 3.2. Figure 5 further corroborates the diurnal periodicity, which was already addressed in Section 4.2. Namely, pronounced peaks in PSD, corresponding to frequencies that contain the most energy and are close to the frequency of 0.0417 h$^{-1}$ (i.e., close to the first mode of the 24 h period), are found

10 for all inspected atmospheric variables (Figures 5b–5e). The same is also found for the lake temperatures in approximately the first 5 meters of the lake (Figure 5a), where as expected, the magnitude of PSD decreased with depth. Notably, higher energies, which are observed for the 24 h period at greater depths (even above 20 m in Figure 5a), do not reflect the general behavior of lake temperatures during the stratification period. Namely, a detailed inspection of separate spectra for individual depths, together with the inspection of shorter time intervals of the observational period (not shown here), revealed that higher energies





at greater depths emerged during stronger wind forcings, while at other times, they were not present. This is in accordance with previous studies pointing to resonant responses of internal seiches modes to diurnal wind forcing (e.g., Antenucci and Imberger, 2003; Vidal et al., 2007; Vidal and Casamitjana, 2008; Simpson et al., 2011; Woolway and Simpson, 2017). In contrast, for depths of $\leq 5$ m, higher energies were found for the 24 h period for both the entire observational period and shorter

time intervals, which indicates typical lake behavior.

Higher modes, which point to an asymmetry in the diurnal variation (e.g., Lundquist and Cayan, 2002), are found for all inspected meteorological variables, and for the uppermost ($\approx 5$ m) lake layer, where the magnitudes of prominent peaks in PSD decreased with the order of a mode (Figure 5a). Furthermore, inspection of individual spectra for each of the investigated greater depths (not shown here) point to a depth of 15 m, where the PSD amplitude for the 3$^{rd}$ mode (period of 8 h) was

noticeably higher than the amplitudes of both the 1$^{st}$ and 2$^{nd}$ modes (namely, $2.6 \cdot 10^{-2}$, $3.0 \cdot 10^{-2}$, and $5.0 \cdot 10^{-2}$ K$^2$s for the 1$^{st}$, 2$^{nd}$ and 3$^{rd}$ modes, respectively). Notably, this depth coincides with the pycnocline/thermocline region or in its vicinity, and as argued above, the diurnal periodicity in the lake temperature at this depth is due to a noncontinuous forcing. Therefore, we hypothesize that the peak for the 8 h period may be associated with internal seiches.

Apart from the diurnal cycle and its higher harmonics, for some variables, other significant periods emerged, such as the air

pressure period of approximately 20 days, which is very likely associated with atmospheric planetary (Rossby) waves (e.g., Pasarić et al., 2000; Šepić et al., 2012). Additionally, a similar period (23 days) also appeared in the lake temperature at a depth of 23 m. However, the possible relationship with atmospheric Rossby waves should be investigated in more detail.



**Figure 5: Power spectral densities (PSD) for lake temperatures at various depths (a), and for the air pressure (b), air temperature (c), wind speed (d), and relative humidity (e) for the entire observational period. All PSDs were computed from the hourly mean values as described in Section 3.2. For meteorological variables, full black lines and shaded areas depict PSDs and 95% confidence intervals, respectively.**

Spectral analysis of the 2-min mean data (Figure 6) revealed prominent peaks in the PSD for periods of approximately 9 min. These peaks are found in the lake temperature time series for depths in the pycnocline/thermocline region (specifically, at depths from 9 to 17 m). An inspection of 2-min mean spectra for individual depths, together with the inspection of shorter time intervals of the observational period (not shown here), revealed that these prominent peaks in PSD emerged due to along-basin

wind forcing, while at other times, they were not present. Furthermore, due to the linear interpolation of frequencies between the two adjacent observational depths, exact peaks for $\approx$ 9 min periods are not as prominent in Figure 6a as they are for individual spectra. As an illustration of these distinguishable peaks seen in the thermocline/pycnocline region for individual depths, we show the results for the depth of 13 m (Figure 6b). Additionally, as seen from the individual spectra (not shown here), prominent peaks are found for the lake temperatures at 17 and 20 m for a period of $\approx$ 6.9 h and for the pycnocline depth

for a period of 49 min. While causes of the periodicities at $\approx$ 6.9 h and 49 min are not clear, periods of $\approx$ 9 min are driven by barotropic surface seiches. Namely, according to the observational study of Kozjak Lake, the principal mode of surface seiches is 9 min (Pasarić and Slaviček, 2016). These barotropic oscillations produce oscillating (upwind-downwind) lake currents, which due to realistic lake basin (that is, due to the inclined lake bottom) oscillate with the same period upslope and downslope. As a result, the thermocline region periodically upwells and downwells. Similar phenomenon, that is, free baroclinic internal

waves produced by barotropic surface seiches is described for the Krka River Estuary, eastern Adriatic coast, Croatia (Orlić et al., 1991).

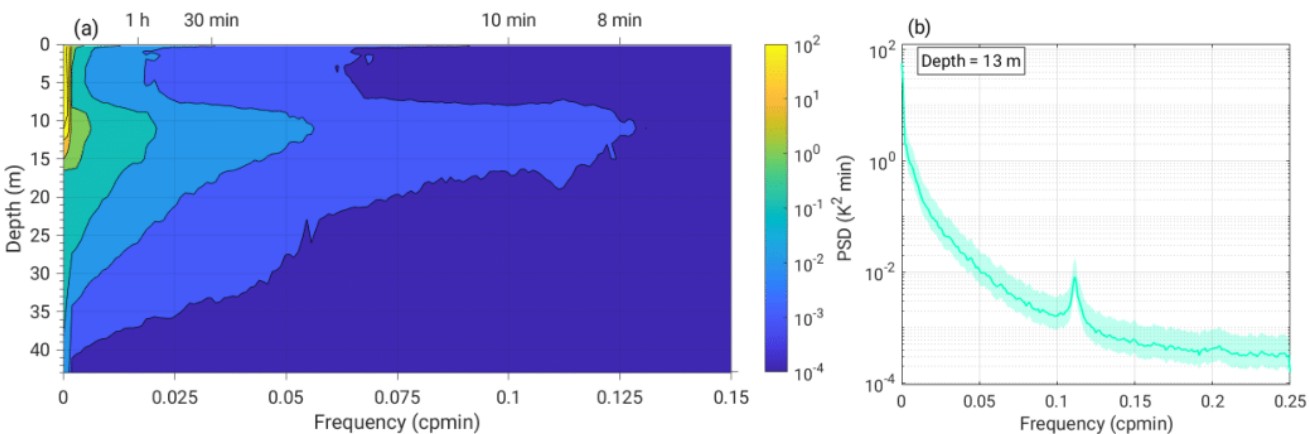

**Figure 6: Power spectral densities computed from the 2-min mean lake temperatures at depths from 0.2 to 43 m (a), and for the**
**depth of 13 m (b) for the entire observational period. PSDs were computed as described in Section 3.2. The full line and shaded area in panel b show the PSD and 95% confidence interval, respectively.**

### 4.4 Episode of internal seiches

As seen from the observed lake temperatures (Figure 2a), oscillations in the lake temperature with amplitudes several times higher than the temperature variations preceding the event emerged at a depth of 15 m on 29 October. Since the depth of 15 m





corresponded to the concurrent position of the thermocline/pycnocline (Figure 3), we will investigate this episode in more detail to determine whether these oscillations were due to internal seiches.

Internal seiches are basin-scale baroclinic standing internal waves that occur due to the presence of two layers of different densities (i.e., epilimnion and hypolimnion) in partially enclosed or enclosed water bodies (Green et al., 1968). They are seen
as periodical changes in the thermocline depth (i.e., as oscillations of the lake temperature at a fixed depth). As with other oscillatory systems, internal seiches form harmonics of higher orders. Internal seiches are always accompanied with surface seiches (e.g., Lemmin et al., 2005). Accordingly, they are generated by the same surface or atmospheric disturbances as surface seiches, such as earthquakes, variable winds, atmospheric pressure disturbances, tides, or heavy precipitation, and winds are considered to be the most important. In the case of a steady wind from one direction, the water level at the opposite (downwind)
end of the basin rises, and the water level lowers at the upwind end. Thus, the epilimnion at the downwind and upwind sides thickens and thins, respectively, while the (initially horizontal) thermocline tilts, that is, downwells at the downwind side and upwells at the upwind side of the lake. When wind stops or suddenly changes direction, an internal seiche is triggered that has a much greater period and amplitude than the accompanying surface seiche. For lakes, due to their typically 1000 times higher amplitudes (e.g., Lemmin et al., 2005; Forcat et al., 2011),  internal seiches are of much greater importance than surface seiches
since they can result in the exchange of the waters between the epilimnion and hypolimnion and energy transfer within the (stratified) lake. Namely, certain events associated with internal seiches (such as disintegration of the basin-scale internal wave into shorter waves, wave breaking, or the occurrence of a large wave amplitude) can produce significant currents and turbulence in the hypolimnon (e.g., Horn et al., 2001; Henderson, 2016). These can lead to both horizontal and vertical transport of heat, water, and nutrients, and can also cause sediment resuspension and affect areal plankton abundance (e.g., Green et al.,
1968; Gaedke and Schimmele, 1991; Kalff, 2002; Lemmin et al., 2005; Stashchuk et al., 2005; Horppila and Niemistö, 2008; Cossu and Wells, 2013).

Watson (1904) was the first to describe internal seiches in lakes. He observed temperature oscillations at fixed depth in Loch Ness, Scotland. The author also applied an idealized two-layer model (Eq. 9) and obtained a period comparable with the observed periods (68 h and ≈ 3 days, respectively). Several decades later, Mortimer (1953) recognized that internal seiches are
free modes of basin oscillations, which emerge as a basin response to episodic wind forcing. Additionally, author argued that in lakes of relatively regular shape ranging from 1.5 to 74 km in length, the 1st (uninodal) mode was always dominant.  Some of the periods of the 1st mode observed by authors worldwide are listed in Table 1. LaZerte (1980) reported the first mode of internal seiches observed in a small lake (Table 1). However, he emphasized that in the case of a small, shallow lake, the metalimnetic (thermocline) region can be as thick or thicker than any of the hypolimnion or epilimnion, and thus, the two-
layer model is not applicable. Furthermore, he argued that the internal seiche structure in that case is much more complex, and that it is dominated by higher vertical modes. Vidal et al. (2007) also pointed to the importance of vertical modes for a deep, warm, monomictic reservoir with a shallow epilimnion and a thick metalimnion (not listed in Table 1).





**Table 1: Observed internal seiches (1st horizontal mode) reported by various authors. TD, TDA and TAS correspond to the reported thermocline depth (m), thermocline depth amplitude (m) and temperature amplitude of a seiche (°C).**

| Lake | Max. lake depth (m) | Max. lake fetch (km) | Period of the 1st mode | Comment | Reference |
|---|---|---|---|---|---|
| Loch Ness, Scotland | 227 | 36.3 | ≈ 3 days | TD = 91 m<br>TAS ≈ 2.8 °C | Watson (1904) |
| Rotoiti, New Zealand | 82 | 15 | 19.6 h | | Green et al. (1968) |
| Frain, US | 10 | 0.5 | 31–41 min | | LaZerte (1980) |
| Baldeggersee, Switzerland | 65 | 4.5 | 7–22 h | TD: 8–22 m<br>TDA (avg.) = 1.5 m | Lemmin (1987) |
| Bodensee (Constance), Germany, Switzerland and Austria | 252 | 64 | 4–6 days | TDA = 12 m | Gaedke and Schimmele (1991) |
| Léman (Geneva), Switzerland and France | 310 | 70.2 | 81.5–136 h | | Lemmin et al. (2005) |
| Lugano, Switzerland and Italy | 288 | | 38 h (southern basin) | TD: 8.6–14.5 m | Hutter et al. (2011) |
| Tanganyka, Tanzania, DRC, Burundi and Zambia | 1470 | 670 | 25–30 days | | Verburg et al. (2011) |
| Simcoe, Canada | 41 | 30 | ≈ 14 h | TD: 12–14 m<br>TDA = 8 m | Cossu and Wells (2013) |
| Alchichica, Mexico | 60 | 1.9 | 12 h | | Filonov et al. (2015) |
| Kozjak, Croatia | 46 | 2.3 | 8.0 h | TD = 15 m<br>TAS = 1.2°C | Present study |
| South Aral Sea, Kazakhstan and Uzbekistan | 37–40 | 175 | 36 h<br>14 h | Weak stratification<br>Strong stratification | Roget et al. (2017) |

5    The particular episode in Kozjak Lake (Figure 7) was initiated on 29 October when oscillations in the lake temperature at the depth of 15 m began. Compared to the amplitudes of temperature variations prior to that date, the amplitudes of these





oscillations were several times higher (Figure 7c). The event was accompanied by a gradual increase in the lake temperature from approximately 9.6°C at the beginning of the episode (29 October at 00:00 LST) to approximately 10.7°C at the end of the observational campaign (5 November at 00:00 LST). Inspection of the wind data shows that from the morning hours of 26 October until the morning hours of 2 November, four southeastern ($\approx$ 135 deg) wind events occurred (Figures 7a and 7b).

Typically, stronger southeastern winds point to a synoptic-scale disturbance, which is over the Adriatic Sea associated with sirocco wind (e.g., Orlić et al., 1994; Horvath et al., 2008; Jeromel et al., 2009; e.g., Pasarić et al., 2009). Simultaneously, over a broader area, the air pressure suddenly drops (Figure 2e), it is cloudy with occasional rain (Figure 2f), and the diurnal course of the air temperature is distorted (Figure 2b). Notably, the southeastern airflow, roughly coincides with the along-basin axis direction (Figure 1, right). Wind speeds in the first two southeastern wind events were higher (maximum speeds $\approx$ 7–8 m s$^{-1}$)

in comparison with the last two events (max. $\approx$ 4 m s$^{-1}$). As seen from the PSDs for lake temperatures during the episode, the 15 m depth stands out with energies 1–3 orders of magnitude higher than the energies obtained for other lake depths (Figure 7d). Prominent peaks are found for the periods of 8.0 h, 4.6 and 2.1 h. A similar pattern (i.e., peaks at 8.0 and 4.6 h), is observed for the 17 m depth, although the peak amplitudes are 3–8 times smaller. For the pycnocline and thermocline depths, elevated energies are associated with periods of 8.0 and 2.1 h (Figure 7h).

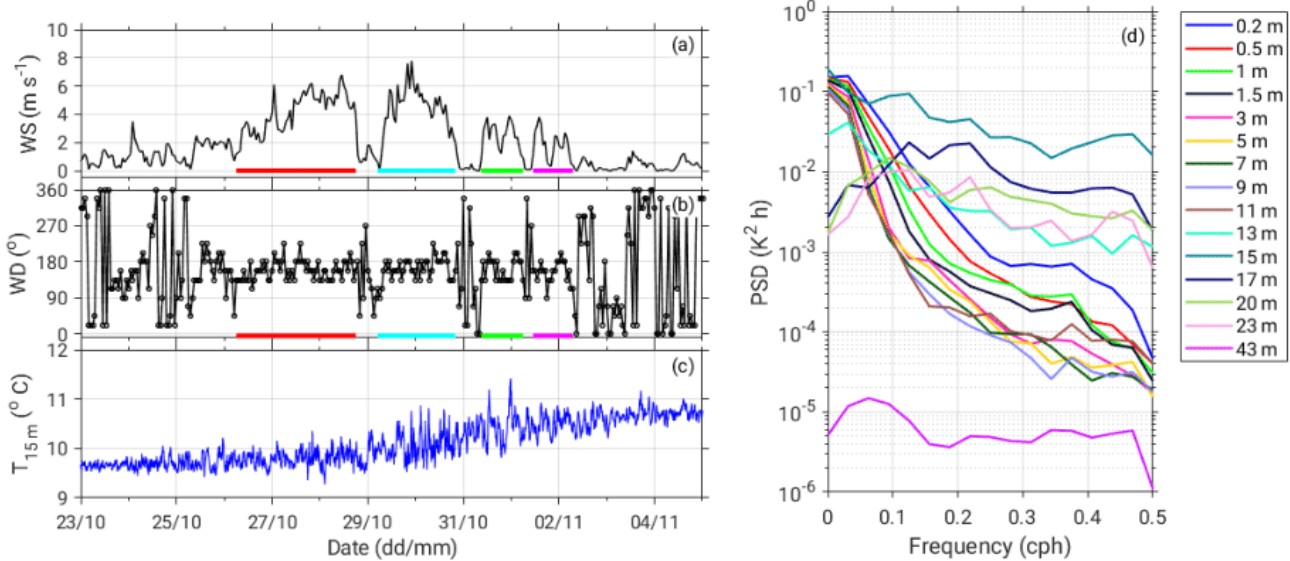

**Figure 7: Observed wind speed (a) and direction (b), and lake temperature at 15 m (c) during 23 October–5 November (both at 00 LST). Four southeastern wind episodes are underlined by different colors (panels a and b). PSDs for the lake temperatures at depths from 0.2 to 43 m (d) and for the pycnocline and thermocline depths (h) are calculated for the period from 28 October to 3 November (both at 00 LST), as described in Section 3.2, except for the window length *WL*. Here, it is set to *WL* = 32 due to the short input time**

**series. Panels e and f show the water density (kg m$^{-3}$) in the epilimnion and hypolimnion during 28 October–3 November, while the period of internal seiches calculated for the idealized rectangular basin (Eq. 9), where *L* = 3095 m, is depicted in panel g.**



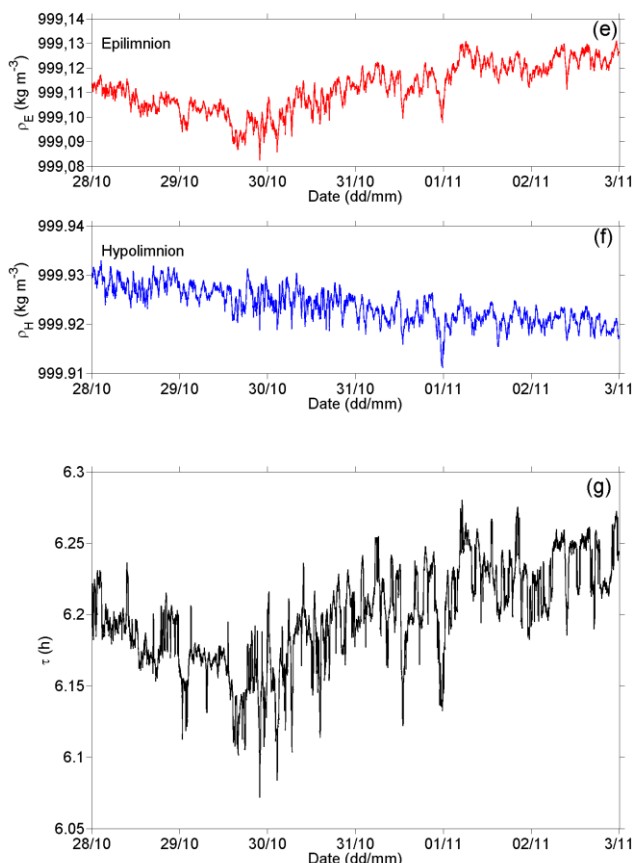

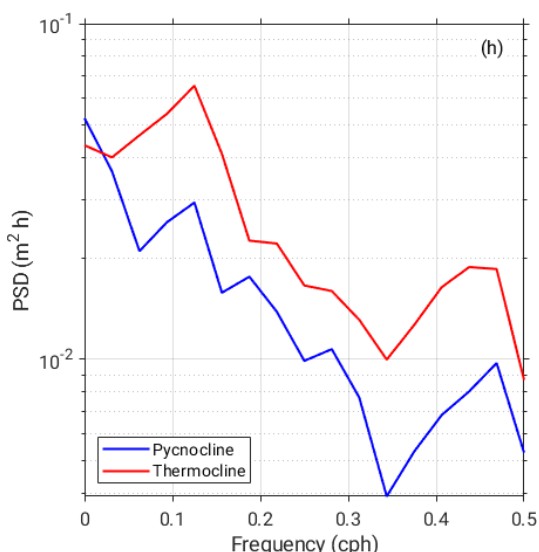

**Figure 7: Cont.**

Figure 7g shows calculated periods of internal seiches for the inspected episode, where a two-layer model of an idealized
rectangular basin was assumed. Under these assumptions, a period of internal seiches can be determined from the following
equation (Watson, 1904):

$$\tau = 2L \ / \ [g(\rho\text{-}\rho\text{'}) \ / \ (\rho/h + \rho\text{'}/h\text{'})]^{1/2}, \tag{9}$$

where $\tau$ is the period (s), $L$ is the basin length (m), $g$ =9.81 m s$^{-2}$ is the acceleration due to gravity, $\rho$ and $\rho\text{'}$ are the densities
of the lower (below the pycnocline) and upper (above the pycnocline) layer (kg m$^{-3}$), respectively, and the corresponding layer
depths are $h$ and $h\text{'}$ (m). As seen from the figure, during the episode, the calculated periods were between 6.07 and 6.24 h,
which is considerably lower than the observed 8.0 h.

Figures 7e and 7f depict the epilimnion and hypolimnion water densities during the episode. These were calculated from the
densities determined at all observational depths from Eq. 5 as the mean densities above and below the pycnocline. As of the





afternoon hours on 29 October, the water density in the epilimnion gradually increased (from approximately 999.09 to approximately 999.13 kg m$^{-3}$ at 3 November at 00 LST), while the water density in the hypolimnion decreased (from approximately 999.93 to 999.92 kg m$^{-3}$). Although these changes are very weak, they still indicate exchange of water between the epilimnion and hypolimnion. Notably, the density change is approximately 4 times higher in the epilimnion than in the

hypolimnion (+0.04 and -0.01 kg m$^{-3}$, respectively), which is due to different volumes of these two layers (for the measuring point, which is shown in Figure 1, the hypolimnion was approximately two times deeper than the epilimnion during the episode).

Finally, we obtained an inverse of Wedderburn number (e.g., Horn et al., 2001; Boegman et al., 2005b) $W^{-1} = 0.059$, where the maximum observed interface (thermocline) displacement and the depth of epilimnion where 0.95 m and 16 m, respectively.

The value of $W^{-1}$, together with a depth ratio (i.e., the ratio of thermocline depth to maximum lake depth, which was for the episode 0.348) points to Regime 1 (damped linear waves, Figure 2 in Horn et al., 2001). This suggests that, the mechanism of degeneration of large-scale internal seiches (that is, the transfer of energy from an initial seiche to smaller scale phenomena) was for investigated episode dominated by viscous damping. In other words, initial basin-scale wave was too small to produce nonlinear phenomena (namely, supercritical flow, shear instabilities or nonlinear steepening and development of solitons).

Instead, the basin-scale lake response to wind forcing was linear. Generally, higher initial wave amplitudes can also produce Regime 1. However, in that case the depth ratio should be close to 0.5, which was not fulfilled for investigated episode. In contrast to nonlinear regimes (Regimes 2–5 in Horn et al., 2001), viscous damping cannot produce high-frequency oscillations. Instead, as illustrated in Figure 7 of Boegman et al. (2005a), it is seen as a damping of the amplitude of the basin-scale internal seiche in time, while the period remains unchanged. This further corroborates our claim that the observed high-frequency

oscillations (Figure 6) were produced by surface seiches.

## 4.5 Multiple linear regression model for near-surface lake temperature

Since the temperature of Kozjak Lake is routinely measured by the Plitvice Lakes National Park at the monthly scale, we aimed to establish a simple model of a near-surface (depth of 0.2 m) lake temperature at a finer temporal resolution. Therefore, we applied multiple linear regression statistical technique (e.g., Dodson, 1992; Bachman et al., 2019), where meteorological

variables, which are routinely observed at the hourly scale, are taken as independent variables. For this purpose we used the procedure ($regres(y,X)$), which is already implemented in MATLAB package.

Prior to the procedure's application, independent variables should be selected based on their individual correlations with dependent variables. As expected, the correlation between the lake temperature at a depth of 20 cm and the individual meteorological variables available (air temperature, wind data, air pressure, precipitation amount, and relative air humidity)

was the highest for the air temperature and wind speed (relative humidity, which also exhibited higher correlation, was eliminated since it strongly depends on the air temperature). However, we note that the concurrent values of any two meteorological variables corresponding to the same measuring point are always at least somewhat correlated. Thus, the





assumption that air temperature and wind speed are mutually independent variables is a rough approximation. Considering this oversimplification, the 2-min mean Kozjak Lake temperature at a depth of 20 cm ($T_{l20}$) can be roughly estimated as follows:

$$T_{l20} = 1.1461 \cdot T_{air} - 0.9411 \cdot v, \tag{10}$$

where $T_{air}$ is the air temperature at 2 m above the ground (°C) and $v$ is the wind speed at 10 m above the ground (m s$^{-1}$). Since
the meteorological variables correspond to the hourly values, their 2-min means were generated through linear interpolation. Figure 8 shows the observed versus modeled near-surface lake temperatures and concurrent values of the air temperature and wind speed. Lake temperatures modeled at a 2-min resolution (Figure 8c) exhibit much higher temporal variations than the observed temperatures. However, the agreement between the observed and modeled temperatures is somewhat improved at a daily resolution (Figure 7d), although the observed temporal variations in the lake temperature are still much smoother than
the modeled variations. The highest discrepancies between the modeled and observed lake temperatures are found after a prominent drop in the air temperature and/or strong winds. Therefore, the above simple, multiple linear regression model can be used as only a crude estimate of daily near-surface (0.2 m) lake temperature in the case of weak winds and undisturbed air temperature patterns. However, we note that the recent employment of a far more complex model, where 20 lake layers are considered (Heiskanen et al., 2015), also resulted in discrepancies between the observed and modeled lake surface temperatures
of several degrees C.

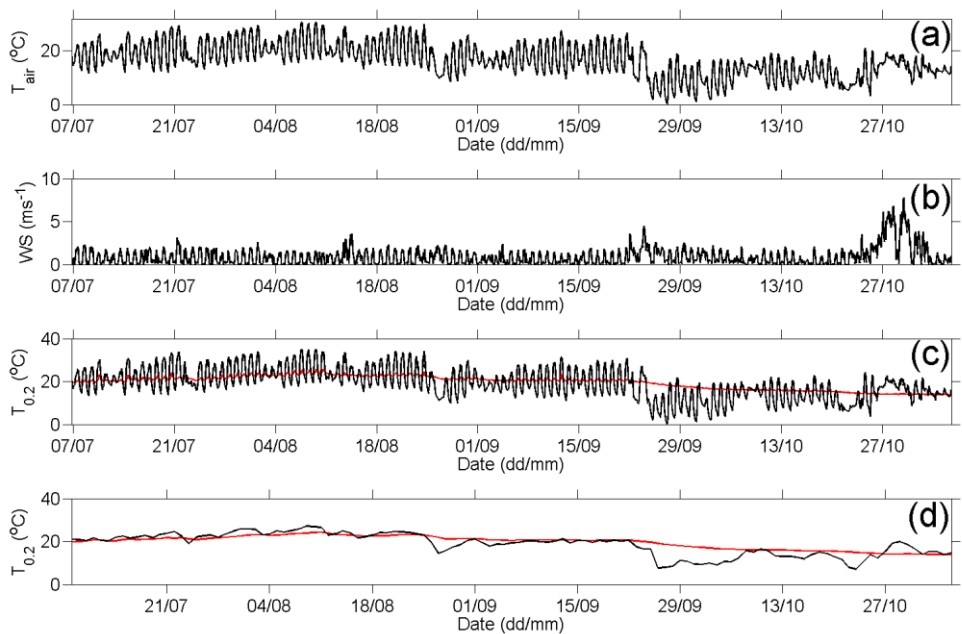

**Figure 8: Observed air temperatures (a) and wind speeds (b). Modeled (black) *vs*. observed (red) lake temperatures at a depth of 0.2 m at hourly (c) and daily (d) resolutions.**



## 5 Conclusions

The aim of the present study was to investigate the fine-scale responses of a stratified, oligotrophic, karstic lake (Kozjak, Plitvice Lakes, Croatia, maximum depth of 46 m and fetch of 2.3 km) to forcings on the lake surface. These responses include pycnocline and thermocline deepening, and the establishment of both forced and free lake temperature oscillations, and

possible water exchange between a hypolimnion and epilimnion. Therefore, we analyzed vertical profiles of lake temperatures observed at resolution of 2 min at 15 depths ranging from near surface to near bottom (i.e., 0.2–43 m) during 6 July–5 November 2018.

During the investigation period, the thermocline deepened from ≈ 10 m (beginning of July) to ≈ 16 m (beginning of November), which corresponds to 3–4 cm per day on average. The maximum observed deepening of the thermocline was approximately

12.5 cm per day and it coincided with the occurrence of internal seiches. The pycnocline followed the same pattern, although it was found approximately 1 m above the thermocline throughout the entire observational period. The highest observed vertical gradients of the water temperature and density in the thermocline and pycnocline regions were $\Delta T/\Delta z$ = -3°C m$^{-1}$ and $\Delta \rho/\Delta z$ = 0.4 kg m$^{-4}$, respectively.

Diurnal variation in lake temperature, which is seen by a naked eye in the first several meters of the lake, is further corroborated

by the results of an hourly data spectral analysis. We conclude that periods associated with the diurnal variation, that is, 24 h and corresponding higher harmonics (12 h, 8 h, 6 h,…, 1 / $n$ h,…, where $n$ = 2, 3, 4,….) correspond to forced oscillations in the lake temperature, which are caused by periodic forcings of heat fluxes on the lake surface. According to the spectral analysis results, these oscillations were present in the first ≈ 5 m of the lake throughout the entire investigation period. The same periods were also observed in all meteorological time series. However, periods of 8.0, 4.6 and 2.1 h, which were found in one single

event for depths of 15 and 17 m and for the pycnocline and thermocline depths, corresponded to the wind-induced baroclinic response of the lake, that is, to internal seiches. These free oscillations in the lake temperatures in the thermocline/pycnocline region initiated approximately 2-3 days after the beginning of the along-the-basin high winds. The oscillations were the most prominent at the depth of thermocline (≈ 15 m), where corresponding PSD amplitude was the highest for 8.0 h. Internal seiches caused the exchange of water between the hypolimnion and epilimnion, which is seen from the simultaneous, slight (~ 0.01

kg m$^{-3}$) increase/decrease in the water density in the epilimnion/hypolimnion.

Forced diurnal oscillations in the lake temperatures (period of 24 h and the higher modes), which are found at greater depths (approximately 7–20 m), were driven by intermittent periodic forcings of stronger winds. Although PSD peaks were seen in the results for the entire observational period, these oscillations were not present in the lake throughout the entire period. Instead, these peaks are a signature of a strong but noncontinuous periodic wind forcing.

To summarize, the results of the present study point to three different types of forcings on the lake surface: (1) the continuous periodic (diurnal) forcing due to heat fluxes; (2) the occasional periodic (diurnal) forcing of stronger winds; and (3) the occasional nonperiodic forcing due to steady, along-the-basin stronger winds. These forcings produced the following lake responses: (1) resulted in continuous forced diurnal oscillations in the lake temperature in the first approximately 5 m thick



layer of the lake; (2) produced occasional forced diurnal oscillations in the lake temperatures at greater depths ($\approx$ 7–20 m); and (3) triggered both free baroclinic oscillations in the thermocline/pycnocline region (i.e., internal seiches) and free barotropic oscillations in the lake surface (surface seiches). Due to realistic-topography conditions, surface seiches produced the oscillating upslope and downslope lake currents. Eventually, these oscillating currents resulted in the free, high-frequency baroclinic oscillations in the thermocline/pycnocline region. Thus, surface seiches should also be considered as possible source of high-frequency internal (baroclinic) oscillations in lakes.

The principal mode of internal seiches (8.0 h) was equal to the 3rd mode of the forced oscillations caused by (2). This resulted in a prominent PSD peak for the lake temperature at 15 m associated with an 8 h period. Accordingly, the resultant energy peak for the 8 h period, which was observed in the spectrum for the entire observational period (although it was caused by the occasional forcings of (2) and (3)), was approximately two times higher than the corresponding peaks in the 1st and 2nd mode. An idealized two-layer model (Eq. 10), suggests a period of internal seiche that is much smaller than the observed 8.0 h. Thus, a two-layer approach is not applicable for a lake basin as complex as Kozjak (a submerged barrier together with the two sub-basins of different depths, an islet, and a departure from an idealized rectangular shape).

In addition to the southeastern wind, a stronger airflow in the wider area of interest can generally be associated with northeastern bora winds over the Adriatic Sea (e.g., Pasarić et al., 2009; Belušić et al., 2013). During the investigation period, several events of northeastern flow occurred, but these events were not accompanied by elevated wind speeds at the meteorological site next to the lake. Therefore, we cannot conclude the possible baroclinic lake response to such flows. Although we anticipate that due to the small northeast-southwest fetch of the lake (which is up to several hundreds of meters at most), the northeastern flow cannot induce internal seiches in Kozjak Lake, this hypothesis should be further investigated.

A spectral analysis of the fine resolution (2-min mean) data for the entire observational period provided insights into the fine scale processes that were caused by the forcing (3) (that is, formation of the free, high-frequency baroclinic waves which were driven by the surface barotropic seiches under realistic-topography conditions). These baroclinic waves were seen in the lake temperature spectra as prominent energy peaks for periods of approximately 9 min in the thermocline region (at depths from 9 to 17 m).

Finally, to obtain a full picture of internal seiches in such a complex basin, a modeling study would be desirable, where the physical processes in the lake could be simulated, for example, by the Semi-implicit Cross-scale Hydroscience Integrated System Model (SCHISM) (Zhang et al., 2016). Furthermore, fine-resolution measurements of the lake temperature profile for both sub-basins would also be needed.

**Acknowledgements.** This study was performed under the project "Hydrodynamic Modeling of Plitvice Lakes System" founded by Plitvice Lakes National Park, Croatia (PLNP). This funding enabled us to purchase the equipment and perform fine-resolution lake-temperature experiments. We also appreciate the technical support of PLNP during equipment installation and data acquisition. Meteorological data were provided by the Croatian Meteorological and Hydrological Service.





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
