# Peer review of "Power spectral densities computed from 2-min mean lake temperatures measured at various depths during 6 July – 5 November 10 2018."

_Hydrology and Earth System Sciences, 2019_

## Referee Comment (RC1) · Anonymous Referee #1 · 17 Feb 2020

In the paper, the authors do a thorough analysis of a single lake in Croatia. They look mostly at observations in the lake and at a meteorological station located on land to the north from the lake observation point. Authors discuss the evolution of thermocline, pycnocline and investigate properties of internal and surface seiches in the lake.

The analysis methods are solid and well explained. It would be easy to reproduce when the data become available, I appreciate the authors' efforts in this regard. I find that the manuscript is well written and I only have a few minor comments and suggestions.

Science: Since the meteorological and lake observation points are not co-located that could introduce uncertainties in the analysis of the causality of forcing on the lake conditions. Have you tried to ensure that those uncertainties are small?

[Figure]

When you were selecting independent variables for the multivariate linear model you have rejected air humidity due to the high correlation with air temperature. What is the correlation value? I am concerned as to how the linear relation (10) eliminates all the high-frequency variability from wind and temperature data. Have you divided the data into fitting and validation parts? Validation should be performed over data that were not used during fitting (i.e. estimation of the parameters). Were the data from the monthly routine measurements by the PLNP used during the parameter estimation for the multivariate model?

Writing:

- P11, L2 (and elsewhere): I would rephrase the "naked eye" because the details seen in the plots are sensitive to the scale you choose. Please try to find a more objective term.

- P16, L18: typo "hypolimnon", missing i.

––––––––––––––––––––––––

---

## Referee Comment (RC2) · Anonymous Referee #2 · 4 Mar 2020

In this study, the physical limnology of a stratified oligotrophic lake was investigated by using observed high-frequency water temperature data monitored at 15 depths. The authors analysed thermocline and pynocline deepening dynamics in accordance with changes in the atmospheric drivers. Further, the impact of three types of meteorological forcings on the lake were investigated: diurnal heat fluxes, noncontinuous strong winds and nonperiodic stronger winds. The authors report that surface waves did generate baroclinic oscillations in the metalimnion under realistic-topography conditions. The study gives interesting insights into the local thermal and fluid dynamics of the study site, which are essential for future modeling projects. Further, the results are interesting (and a helpful resource) for limnologists, meteorologists and civil engineers. The methods and results are well explained, although some paragraphs seems unnec-

essary for the overall study.

General points:

- the last paragraph about the multiple linear regression of water temperatures at 0.2 m depth and meteorological drivers feels - in my opinion - unnecessary for the paper, as the results seem rather weak and were not used in the study at all. I would suggest just removing it and maybe including it in a follow-up work to make the current paper more coherent.

- I wanted to say that I really liked reading the thorough descriptions of the 'thermocline and pyncocline' and 'spectral analysis' paragraphs

- Do you feel the general data supports the detailed investigation of the thermal regime? Most loggers in the suggested thermocline region have a spatial distance of 3 m, which was then linearly interpolated to spacings of 0.25 m. Adding a section in which these methodological uncertainties are discussed could benefit the reader in my opinion (this uncertainty is only briefly mentioned at P15). Or were the linearly interpolated data only used for thermocline and pynocline calculations?

- Just a suggestion, but I would keep units as e.g. kg (m-3 m-1) instead of kg m-4 to clarify the gradient

- Dates should be Month Day not Day Month, e.g. P22 L 6

Additional points:

- P2, L20: 'Some authors' is a very vague statement. Although you go into more detail in the next sentences, you could rephrase it to e.g. 'There are several studies reporting observed high-...'

- P2, L 23: Is 'Authors' the link to Thorpe et al.? Otherwise this is unclear.

- P4, 'Lake temperatures': Could you also please state the resolution of the thermistors.

- P4, 'Meteorological data': Although the meteorological station is close (less than 1.6 km away, right?), did you check for uncertainties when using the meteorological data for the interpretation of the buoy data?

- P6, L16: Maybe I'm mistaken but as Welch's method is some kind of overlapping windowed Fourier transform method, is the sentence "Therefore, the Fourier transform computation is not applicable" a bit misleading?

- P7, L21: I would substitute 'naked eye' with a more meaningful term

- P9, L 14: Why were such high gradients expected for a Mediterranean lake? It's quite hard to see the gradients of -7 deg C/m in the contour plots. As this would correspond to a temp decrease of 1.75 deg C over 0.25 m, I guess such high gradients could only be observed in Mid August?

- P9, L28: Out of curiosity, what's the reason for including both thermocline and pycnocline in this study? You're stating that both were calculated from temperature data and due to the non-linear calculation of density from water temp in freshwater lakes, they do not coincide. Still, I think the implementations of differences between both boundaries aren't discussed in the manuscript. Would just stating/showing either thermocline or pycnocline also be enough for the purpose of this study?

- P10, L11-12: Personally I would delete the sentences "The results for N2 . . . . This result is expected .. of water density." as you are not showing these results and you're mostly stating the obvious for the buoyancy frequency.

- P10, L15: Does "4-16*10-3" mean N2 was between 4*10-3 and 16*10-3 s-2?

- P11, L13: Is the occurrence of this cold water parcel a proof for the favorable vertical mixing conditions? And is this cold water parcel 'real' or a just an artifact from the averaging to a daily contour plot? Or, are most days during summer showing these daily dynamics, or is this just because some days in November have the phenomenon?

- P11, L19: Again, I would delete the sentence about the N2 daily course as it's not

shown and has, as expected, the same pattern as the density gradient.

- P12, L9: Could you please add a vertical line for the 0.0417 h-1 frequency in the plots 5b-e, that would help identifying the first mode more easily.

- P12, L14: The detailed inspections of the spectra at greater depths sound interesting. I think a reference here to the supplementary material is missing (in which the N2 plots could also be added if necessary)

- P13, L6-8:The whole sentence "Higher modes, ...of a mode (Figure 5a)" is unclear to me.

- P13, L8-13: Isn't this a very important paragraph for the whole study by stating that you found a significantly higher 3rd mode amplitude at 15 m depths with a period of 8 h? I would argue in showing this figure in the manuscript. Why is the PSD for 13 m depth in the manuscript, but not for depth 15 m? Also I would not use "namely" and "respectively" in the same sentence. Further, has Fig 6 b the right unit in the y axis, as the same figure in the supplementary information has 'K2 s'

- P15, L13: Could you please explain in more detail what 'realistic lake basin conditions' means and why these currents oscillate with the same period upslope and downslope?

- P16, L 8: You're stating winds as important drivers twice in this sentence

- P16, L22-32: This whole paragraphs feels like it could either be cut or that it should be in the introduction and not in the results/discussion paragraph

- P19, L4: Can the two-layer model assuming a rectangular basin be used for this inclined lake with the barrier separating two lake basins?

- P20, L8: I would suggest showing the Wedderburn equation in the manuscript

- P20, L10-L20: This paragraph is hard to understand when not simultaneously reading Horn et al. I like how it connects the discussion with the introduction, but could you please give more information regarding the findings of Horn (2001) and Boegman

(2015) without having the reader refer to the specific figures.

- P23, L7: To avoid confusion, could you please exchange (2) here with 'occasional periodic forcing of stronger winds' as otherwise it could be confused with the other (2) which is 'produced occasional forced diurnal circulations'. The same is true for L10 and L 21 on P23

- P23, L13: I think the discussion of the unsuitable two-layer model should happen before the conclusions paragraph.

---

## Author Comment (AC2) · 17 Mar 2020

Comment: Writing: - P11, L2 (and elsewhere): I would rephrase the "naked eye" because the details seen in the plots are sensitive to the scale you choose. Please try to find a more objective term. - P16, L18: typo "hypolimnon", missing i.

Response: The "naked eye" is removed and the typo is corrected.

---

## Author Response (AR1)

First of all, we would like to thank both Referees for very useful comments. Our response is as follows.

Referee #1
**Comment:** In the paper, the authors do a thorough analysis of a single lake in Croatia. They look mostly at observations in the lake and at a meteorological station located on land to the north from the lake observation point. Authors discuss the evolution of thermocline, pycnocline and investigate properties of internal and surface seiches in the lake. The analysis methods are solid and well explained. It would be easy to reproduce when the data become available, I appreciate the authors' efforts in this regard. I find that the manuscript is well written and I only have a few minor comments and suggestions. Science: Since the meteorological and lake observation points are not co-located that could introduce uncertainties in the analysis of the causality of forcing on the lake conditions. Have you tried to ensure that those uncertainties are small?

**Response:** Regrettably, we do not have co-located meteorological and lake-temperature data to assess the possible uncertainties. However, the lake and meteorological measuring sites are very close to each other (distance≈1.6 km, Figure 1, right).Generally, we would expect larger differences in meteorological conditions between the two sites if they were separated by one or more topographic obstacles. Topographic obstacle(s) may affect local meteorological conditions due to up- and down-slope winds, blocking of the airflow and other influences. Here, the meteorological site is positioned on the first slope next to the lake and there are no topographical obstacles between the two sites. Thus, we assume that meteorological conditions at the two sites are very similar. Some small differences in meteorological conditions may occur, such as slightly stronger winds over the lake in comparison with winds above the ground (due to the weaker surface friction) or slightly lower/higher air temperature above the lake during the day/night (due to different heat capacity of water and soil). Nevertheless, main characteristics of meteorological forcing, as are, for example, diurnal periodicity of both the air temperature and wind speed and strength of the airflow, which are important forthe present study, are expected to be very similar at the two sites.

**Comment:** When you were selecting independent variables for the multivariate linear model you have rejected air humidity due to the high correlation with air temperature. What is the correlation value? I am concerned as to how the linear relation (10) eliminates all the high-frequency variability from wind and temperature data. Have you divided the data into fitting and validation parts? Validation should be performed over data that were not used during fitting (i.e., estimation of the parameters). Were the data from the monthly routine measurements by the PLNP used during the parameter estimation for the multivariate model?

**Response**: Based on Referee #2's suggestion, we decided to remove the section 4.5 Multiple linear regression model for near-surface lake temperature from the revised manuscript. Nevertheless, we would like to answer Referee #1's questions regarding multivariate linear model. Yes, we rejected the air humidity due to the high correlation with the air temperature. Correlation coefficient between the air temperature and the relative humidity (calculated from 2920 pairs of hourly data) is R = -0.63. Thank you for drawing our attention to the importance of data division into fitting and validation parts. In the previous manuscript version we did not divide the data. In the meantime, the new dataset (summer 2019) became available, so we validated the model based on the new data. Results are shown in the Fig. 1 of the present response. Finally, monthly routine measurements of the lake temperature performed by the Plitvice Lakes National Park (PLNP) where not used while building the multivariate model. Model coefficients were determined on the basis of hourly meteorological data and hourly lake temperature data, the latter being calculated from 2-min values obtained in the framework of a temporary measurement program.

[Figure]

**Fig. 1.**Observed air temperatures (a) and wind speeds (b). Modeled (black) vs. observed (red) lake temperatures (0.2 m) at hourly (c) and daily (d) resolutions for validation dataset (7 Jul – 6Nov 2019).

**Comment:** Writing: - P11, L2 (and elsewhere): I would rephrase the "naked eye" be-cause the details seen in the plots are sensitive to the scale you choose. Please try tofind a more objective term. - P16, L18: typo "hypolimnon", missing i

**Response:** The "naked eye" is removed and the typo is corrected.

**Referee #2**
**Comment:** General points: - the last paragraph about the multiple linear regression of water temperatures at 0.2 m depth and meteorological drivers feels - in my opinion - unnecessary for the paper, as the results seem rather weak and were not used in the study at all. I would suggest just removing it and maybe including it in a follow-up work to make the current paper more coherent.

**Response:** We removed the section 4.5 Multiple linear regression model for near-surface lake temperature from the revised manuscript.

**Comment:** - I wanted to say that I really liked reading the thorough descriptions of the 'thermocline and pyncocline' and 'spectral analysis' paragraphs.
- Do you feel the general data supports the detailed investigation of the thermal regime?
Most loggers in the suggested thermocline region have a spatial distance of 3 m, which was then linearly interpolated to spacings of 0.25 m. Adding a section in which these methodological uncertainties are discussed could benefit the reader in my opinion (this uncertainty is only briefly

mentioned at P15). Or were the linearly interpolated data only used for thermocline and pynocline calculations?

**Response**: Vertical temperature and density profiles suggest that the Nyquist wavelength is approximately 10–12 m, implying that the vertical profiles are adequately captured with our vertical sampling rates of 2-3 m. This further means that linear interpolation onto the 0.25 m vertical grid is acceptable and, anyhow, the interpolation is used only while calculating the thermocline and pycnocline depths.

**Comment:** Just a suggestion, but I would keep units as e.g. kg (m-3 m-1) instead of kg m-4 to clarify the gradient.

**Response:** $kg\ m^{-4}$ is replaced by $(kg\ m^{-3})\ m^{-1}$ throughout the entire text.

**Comment:** Dates should be Month Day not Day Month, e.g. P22 L 6

**Response:** According to the manuscript preparation guidelines https://www.hydrology-and-earth-system-sciences.net/for_authors/manuscript_preparation.html, the date should be given in format Day Month Year. Therefore, we have kept the present date system.

**Comment:** Additional points: - P2, L20: 'Some authors' is a very vague statement. Although you go into more detail in the next sentences, you could rephrase it to e.g. 'There are several studies reporting observed high-...'

**Response:** The text is changed as suggested by the Referee.

**Comment:** P2, L 23: Is 'Authors' the link to Thorpe et al.? Otherwise this is unclear.

**Response:** Yes, thank you. We corrected the text. 'Authors' are replaced with 'The authors'.

**Comment:** P4, 'Lake temperatures': Could you also please state the resolution of the thermistors.

**Response:** Temporal resolution is described in the following sentences: "The sensors measure temperature every second, while the averaging interval of the stored data is specified by a user. In the present study we stored the 2-min means.", whereas the spatial distribution is specified in the following sentence: "Fifteen factory calibrated sensors were fastened to a string at fixed depths ranging from 0.2 to 43 m (specifically, at 0.2, 0.5, 1, 1.5, 3, 5, 7, 9, 11, 13, 15, 17, 20, 23, and 43 m). The string was attached to a buoy that was moored to ensure its fixed position in the deepest part (46 m) of the lake ($\varphi$ = 44.8902°N, $\lambda$ = 15.6038°E; Figure 1, right)."

**Comment:** 'Meteorological data': Although the meteorological station is close (less than 1.6 km away, right?), did you check for uncertainties when using the meteorological data for the interpretation of the buoy data?

**Response**: Please see our response to the first comment of Referee #1.

**Comment:** P6, L16: Maybe I'm mistaken but as Welch's method is some kind of overlapping windowed Fourier transform method, is the sentence "Therefore, the Fourier transform computation is not applicable" a bit misleading?

**Response:** Thank you for pointing to the vagueness of this sentence. Welch's method is an overlapping windowed Fourier transform method. What we meant here is that the straightforward

Fourier transform method is inapplicable (Solomon Jr., 1991, in References of the present study). The revised text is corrected accordingly.

**Comment:** P7, L21: I would substitute 'naked eye' with a more meaningful term

**Response**: The "naked eye" is removed.

**Comment:** P9, L 14: Why were such high gradients expected for a Mediterranean lake? It's quite hard to see the gradients of -7 deg C/m in the contour plots. As this would correspond to a temp decrease of 1.75 deg C over 0.25 m, I guess such high gradients could only be observed in Mid August?

**Response**: Although the Kozjak Lake (535 m above the sea level) is geographically close to the Adriatic Sea (approximately 50 km distant), climate conditions in the lake area are not quite Mediterranean. Namely, the longest and one of the highest Croatian mountains (Velebit Mountain, 145 km long, 1757 m high), which stretches along the Adriatic coast,  separates coastal areas (with Mediterranean climate) from the inland area (where the lake is located). In addition, the lake itself is in a mountainous region, between the mountains 1280, 1640 and 884 m high (e.g., Babinka, 2007, in References).  Previous climatological studies of the greater lake area Makjanić (1958; 1971–1972), and the study of stable isotopes of oxygen and hydrogen in precipitation over Croatia (Hunjak, 2015), show that the lake area is at the border between the maritime and continental climate regions. Specifically, the lake area is at the border between two climatic zones as defined by the Köppen climate classification (e.g., Kottek et al., 2006): the temperate climate zone (C) and the snow climate zone (D).

And yes, it is difficult to see such high gradients in the contour plots showing entire lake depth. However, the high gradients are not so rare in the uppermost part of the lake from July to mid-September. For the information, in Figure 1 of this response we show gradients for the first three layers throughout the entire period.

[Figure]

**Figure 1.** Vertical gradients of water temperature for the first three layers (0.2–0.5, 0.5–1.0 and 1.0–1.5 m).

Additional references:

Hunjak, T. (2015): Prostorna distribucija stabilnih izotopa kisika i vodika iz oborine na području Republike Hrvatske. Doktorska disertacija, Sveučilište u Zagrebu, Prirodoslovno-matematički fakultet, Zagreb, 82 pp.

Kottek, M., Grieser, J., Beck, C., Rudolf, B. and Rubel, F. (2006): World map of the Köppen-Geiger climate classification updated, Meteorol. Z., 15, 259–263, DOI: 10.1127/0941-2948/2006/0130.

Makjanić, B. (1958): Prilog klimatografiji područja Plitvičkih jezera, u Nacionalni park Plitvička jezera, Josip Šafar (urednik), Poljoprivredni nakladni zavod, Zagreb, 357–390.

Makjanić, B. (1971–1972): O klimi užeg područja Plitvičkih jezera, Geografski glasnik, 33–34, 5–24.

(Results of the above studies published in Croatian are summarized by Klaić et al. (2018)).

**Comment:** Out of curiosity, what's the reason for including both thermocline and pycnocline in this study? You're stating that both were calculated from temperature data and due to the non-linear calculation of density from water temp in freshwater lakes, they do not coincide. Still, I think the implementations of differences between both boundaries aren't discussed in the manuscript. Would just stating/showing either thermocline or pycnocline also be enough for the purpose of this study?

**Response:** Both temperature and density profiles may be encountered in the literature and by considering both of them we wanted to draw readers' attention to possible differences between the results – as, for example, thermocline and pycnocline depths – stemming from the two.

**Comment:** P10, L11-12: Personally I would delete the sentences "The results for N2 . . . . This result is expected .. of water density." as you are not showing these results and you're mostly stating the obvious for the buoyancy frequency.

**Response**: These sentences are deleted.

**Comment:** P10, L15: Does "4-16*10-3" mean N2 was between 4*10-3 and 16*10-3 s-2?

**Response:** Yes, it does. The text is corrected accordingly.

**Comment:** P11, L13: Is the occurrence of this cold water parcel a proof for the favorable vertical mixing conditions? And is this cold water parcel 'real' or a just an artifact from the averaging to a daily contour plot? Or, are most days during summer showing these daily dynamics, or is this just because some days in November have the phenomenon?

**Response**: No, for the times between 5 and 7 LT (close to the sunrise), a large number of profiles exhibit such behavior. That is, at depths between 0.2 and 1.5 m the water is frequently somewhat colder than the water below. On the contrary, such pattern is not so frequent at other times of the day. As an illustration, Figure 2 of this response shows diurnal variation of absolute frequency of positive vertical temperature gradient within the layer between 1 and 1.5 m. Positive gradient means that within this layer the lake temperature increases with depth, thus producing conditions favorable for vertical mixing. As seen from the figure, for July, August and entire dataset such conditions were most frequent between 4 and 5 LT, for September they were most frequent for 3–4 and 5–6 LT, while for October they were most frequent for 3–6 LT. (November is not shown, since there where only a few November days with lake temperature measurements.) On the contrary, during the daytime, from 8 LT to approximately 18–20 LT (depending on particular month), absolute frequencies of

positive vertical temperature gradient are substantially lower than those around the sunrise. For September and October, high absolute frequencies (comparable with the "sunrise" frequencies) are also found for nighttime hours.

[Figure]

**Figure 2.** Diurnal variation of absolute frequency of observations with positive vertical temperature gradient (conditions for vertical mixing of the lake water) within the layer between 1 and 1.5 m.

**Comment:** P11, L19: Again, I would delete the sentence about the N2 daily course as it's not shown and has, as expected, the same pattern as the density gradient.

**Response**: The two sentences regarding N2 are deleted.

**Comment:** P12, L9: Could you please add a vertical line for the 0.0417 h-1 frequency in the plots 5b-e, that would help identifying the first mode more easily.

**Response**: Vertical lines in updated Fig.5b-f are added.

**Comment:** P12, L14: The detailed inspections of the spectra at greater depths sound interesting. I think a reference here to the supplementary material is missing (in which the N2 plots could also be added if necessary).

**Response**: The supplementary material with hourly spectra at individual depths for both the entire observational interval and the interval without strong winds is added (SUPPLEMENT 1). For the entire interval (left panels), 24 h period is prominent at greater depths (from 7 to 23 m), while the same is not found for the interval characterized by weak winds (1–20 September, right panels).

**Comment**: P13, L6-8: The whole sentence "Higher modes, ...of a mode (Figure 5a)" is unclear to me.

**Response:** Thank you for pointing to the error. The "modes" are replaced by the "harmonics".

**Comment**: P13, L8-13: Isn't this a very important paragraph for the whole study by stating that you found a significantly higher 3rd mode amplitude at 15 m depths with a period of 8 h? I would argue in showing this figure in the manuscript. Why is the PSD for 13 m depth in the manuscript, but not for depth 15 m? Also I would not use "namely" and "respectively" in the same sentence. Further, has Fig 6 b the right unit in the y axis, as the same figure in the supplementary information has 'K2 s'

**Response:** Hourly results for 15 m are now shown in Figure 5b and the text is modified accordingly. The sentence with "namely" and "respectively" is now rewritten. The error in units is corrected in supplementary material (SUPPLEMENT 2). Correct units are as in Figure 6b, that is "K$^2$min". Also, in the current Figure 6 we replaced results for 13 m by results for 15 m.

**Comment**: P15, L13: Could you please explain in more detail what 'realistic lake basin conditions' means and why these currents oscillate with the same period upslope and downslope?

**Response**: The text is now rewritten as follows: "These barotropic oscillations of the lake surface produce oscillating (upwind-downwind) lake currents which have the same period as oscillations of the lake surface. In an idealized case (over a flat lake bottom) these currents would be horizontal. However, in the realistic lake basin (that is, the basin with the inclined bottom) the currents are forced to oscillate upslope and downslope."

**Comment**: P16, L 8: You're stating winds as important drivers twice in this sentence.

**Response:** The text is rewritten as follows:" Accordingly, they are generated by the same surface or atmospheric disturbances as surface seiches, such as earthquakes, variable winds, atmospheric pressure disturbances, tides, or heavy precipitation, with the winds being considered to be the most important driving agents."

**Comment:** L22-32: This whole paragraphs feels like it could either be cut or that it should be in the introduction and not in the results/discussion paragraph.

**Response**: Although in this paragraph the past studies of internal seiches are reviewed, we decided to keep it here since this Section is devoted solely to internal seiches. Thus, the reader can more easily (without scrolling through the text) compare our lake characteristics and results with those given by other authors.

**Comment:** P19, L4: Can the two-layer model assuming a rectangular basin be used for this inclined lake with the barrier separating two lake basins?

**Response**: Obviously not. This is now stated in both the paragraph following equation (9): "As seen from the figure, during the episode the calculated periods were between 6.07 and 6.24 h, which is considerably lower than the observed 8.0 h. Thus, we conclude that the idealized two-layer model is not suitable for estimation of the period of internal seiche in Kozjak Lake." and in the Conclusions: "An idealized two-layer model (Eq. 9) suggests a period of internal seiche that is much smaller than the observed 8.0 h. Thus, a two-layer approach is not applicable for estimation of the period of internal seiche for a lake basin as complex as Kozjak (which includes submerged barrier together with the two sub-basins of different depths and an islet, and therefore considerably departs from an idealized rectangular shape)."

**Comment**: P20, L8: I would suggest showing the Wedderburn equation in the manuscript.

**Response**: The formula for Wedderburn number ($W$) is now added, as $W^{-1} = z_{max} / h_e$, where $z_{max}$ is the amplitude of the initial disturbance and $h_e$ is the depth of the epilimnion (new Equation (10)).

**Comment:** P20, L10-L20: This paragraph is hard to understand when not simultaneously reading Horn et al. I like how it connects the discussion with the introduction, but could you please give more information regarding the findings of Horn (2001) and Boegman without having the reader refer to the specific figures.

**Response**: Figure 2 of Horn et al. (2001) (which is also published in Boegman et al. (2005a) as Fig. 1) is derived from laboratory experiments and field data. It defines separate regimes based on the values of inverse Wedderburn number and the ratio $h_e/H$, where $H$ is the maximum lake depth. We find it quite difficult to describe several nonlinear curves that separate regimes one from the other, and, regrettably, we do not have permission to reproduce the figure. However, if not otherwise, Boegman et al. (2005a) is easily accessible via ResearchGate.

**Comment:** P23, L7: To avoid confusion, could you please exchange (2) here with 'occasional periodic forcing of stronger winds' as otherwise it could be confused with the other (2) which is 'produced occasional forced diurnal circulations'. The same is true for L10 and L 21 on P23.

**Response**: The text is corrected as suggested by the Reviewer.

**Comment**: P23, L13: I think the discussion of the unsuitable two-layer model should happen before the conclusions paragraph.

**Response**: Unsuitability of the two-layer model is additionally emphasized by the following: "Thus, we conclude that the idealized two-layer model is not suitable for estimation of the period of internal seiche in Kozjak Lake." (Section 4.4)

[revised manuscript text omitted]
), at the depth of 15 m (b) and for the air pressure (bc), air temperature (cd), wind speed (de), and relative humidity (ef) for the entire observational period. All PSDs were computed from the hourly mean values as described in Section 3.2. For meteorological variables, full blackFull lines and shaded areas in panels b–f depict PSDs and 95% confidence intervals, respectively., whereas vertical dashed lines indicate the 24 h period.**

Spectral analysis of the 2-min mean data (Figure 6) revealed prominent peaks in the PSD for periods of approximately 9 min. These peaks are found in the lake temperature time series for depths in the pycnocline/thermocline region (specifically, at depths extending from 9 to 17 m). An inspection of 2-min mean spectra for individual depths, together with the inspection of shorter time intervals of the observational period (not shown here), revealed that these prominent peaks in PSD emerged due to along-basin wind forcing, while at other times, they were not present. Furthermore, due to the linear

interpolation of frequencies between the two adjacent observational depths, exact peaks for ≈ 9 min periods are not as prominent in Figure 6a as they are for individual spectra. As an illustration of these distinguishable peaks seen in the thermocline/pycnocline region for individual depths, we show the results for the depth of 15 m (Figure 6b). Additionally, as seen from the individual spectra (not shown here), prominent peaks are found for the lake temperatures at 17 and 20 m for a period of ≈ 6.9 h and for the pycnocline depth for a period of 49 min. While causes of the periodicities at ≈ 6.9 h and 49 min are not clear, periods of ≈ 9 min are driven by barotropic surface seiches. Namely, according to the observational study of Kozjak Lake, the principal mode of surface seiches is 9 min (Pasarić and Slaviček, 2016). These barotropic oscillations of the lake surface produce oscillating (upwind-downwind) lake currents which  have the same period as oscillations of the lake surface. In an idealized case (over a flat lake bottom) these currents would be horizontal. However, in the realistic lake basin (that is, the basin with the inclined  bottom) the currents are forced to oscillate  
[revised manuscript text omitted]

Figures 7e and 7f depict the epilimnion and hypolimnion water densities during the episode. These were calculated from the densities determined at all observational depths from Eq. 5 as the mean densities above and below the pycnocline. As of the afternoon hours on 29 October, the water density in the epilimnion gradually increased (from approximately 999.09 to approximately 999.13 kg m$^{-3}$ at 3 November at 00 LST), while the water density in the hypolimnion decreased (from approximately 999.93 to 999.92 kg m$^{-3}$). Although these changes are very weak, they still indicate exchange of water between the epilimnion and hypolimnion. Notably, the density change is approximately 4four times higher in the epilimnion than in the hypolimnion (+0.04 and -0.01 kg m$^{-3}$, respectively), which is due to different volumes of these two layers (for the measuring point, which is

shown in Figure 1, the hypolimnion was approximately two times deeper than the epilimnion during the episode).

Finally, we  employed Wedderburn formula (e.g., Horn et al., 2001; Boegman et al., 2005b):

$$W^{-1} = z_{max} / h_e, \hspace{8em} (10)$$

where $z_{max}$ is the amplitude of the initial disturbance (i.e., the maximum interface displacement) and $h_e$ is the depth of epilimnion. For the maximum observed interface (thermocline) displacement of 0.95 m and the depth of epilimnion of 16 m we obtained an inverse of Wedderburn number $W^{-1} = 0.059$. 
[revised manuscript text omitted]
 period of 3$^{rd}$ harmonics of  forced oscillations caused by (occasional periodic forcing of stronger winds

(process 2 indicated above). This resulted in a prominent PSD peak for the lake temperature at 15 m associated with an 8 h period. Accordingly, the resultant energy peak for the 8 h period, which was observed in the spectrum for the entire observational period (although it was caused by the occasional forcing of strong periodic (2) and strong along-the-basin nonperiodic (3) winds), was approximately two times higher than peaks corresponding to the 1$^{st}$ and 2$^{nd}$ harmonics.

An idealized two-layer model (Eq. 9) suggests a period of internal seiche that is much smaller than the observed 8.0 h. Thus, a two-layer approach is not applicable for estimation of the period of internal seiche for a lake basin as complex as Kozjak (which includes submerged barrier together with the two sub-basins of different depths and an islet, and therefore considerably departs from an idealized rectangular shape).

In addition to the southeastern wind, a stronger airflow in the wider area of interest can generally be associated with northeastern bora winds over the Adriatic Sea (e.g., Pasarić et al., 2009;  Belušić et al., 2013). During the investigation period, several events of northeastern flow occurred, but these events were not accompanied by elevated wind speeds at the meteorological site next to the lake. Therefore, we cannot conclude on the possible baroclinic lake response to such flows. Although we anticipate that due to the small northeast-southwest fetch of the lake (which is up to several hundreds of meters at most), the northeastern flow cannot induce internal seiches in Kozjak Lake; this hypothesis should be further investigated.

A spectral analysis of the fine resolution (2-min mean) data for the entire observational period provided insights into the fine scale processes that were caused by the forcing of the strong, steady, along-the-basin winds (3) (that is, formation of the free, high-frequency baroclinic waves which were driven by the surface barotropic seiches under realistic-topography conditions). These baroclinic waves were seen in the lake temperature spectra as prominent energy peaks for the periods of approximately 9 min in the thermocline region (at depths extending from 9 to 17 m).

Finally, to obtain a full picture of internal seiches in such a complex basin, a modeling study would be desirable, where the physical processes in the lake could be simulated, for example, by the Semi-implicit Cross-scale Hydroscience Integrated System Model (SCHISM) (Zhang et al., 2016). Furthermore, fine-resolution measurements of the lake temperature profile for both sub-basins would also be useful.

*Data availability*. Lake temperature data are available on request.

*Author contributions*. ZBK designed and performed the experiment and wrote the paper. KB performed spectral analyses and produced majority of figures. MO interpreted spectral analyses results and contributed to conclusions.

*Competing interests*. The authors declare that they have no conflict of interest.

**Acknowledgements.** Very useful comments of the two anonymous referees are greatly appreciated. 
[revised manuscript text omitted]